# Identifying Water Crossings in Rural Liberia and Rwanda Using Remote and Field-Based Methods

**Kyle Shirley [1], Abbie Noriega [1], Davey Levin [1] and Christina Barstow [1,2,*]**

[1] Bridges to Prosperity, 1031 33rd Street, Denver, CO 80205, USA; kyleshirley@bridgestoprosperity.org (K.S.); abbienoriega@bridgestoprosperity.org (A.N.); sbura.umoya@gmail.com (D.L.)

[2] Mortenson Center in Global Engineering, University of Colorado, Boulder, CO 80303, USA

\* Correspondence: barstow@colorado.edu

**Abstract:** Safe and consistent access to essential services is critical for poverty alleviation in rural communities, but even significant physical transportation barriers, such as pedestrian water crossings, are poorly mapped, leaving the scope of need for rural trailbridges largely unknown. Field-based efforts to catalogue those barriers can be effective but are costly and time-consuming. The study described here details field-based methods for identifying pedestrian water crossings in rural Liberia and Rwanda, as well as remote methods, to evaluate their effectiveness and potential application for assessing future rural infrastructure networks. The work highlights challenges, addresses components of the field-based method that limit scalability on a global level, and outlines a way forward for future endeavors to identify pedestrian water crossings. Overall, the most effective remote method applied in this study identified 16 percent of the crossings identified using field-based methods in the same area of interest in Liberia, and 72 percent of the crossings identified using field-based methods in the same area of interest in Rwanda. The field-based method remains the most effective method for bridge site identification, though the significant resources required for an effective field study underscore the need for greater investment in remote methods. Additionally, as neither method alone yields results that fully encapsulate bridge need, the authors recommend a blended approach that incorporates a more sophisticated remote method with streamlined field-based methods that leverage existing local knowledge and expertise.

**Keywords:** rural infrastructure; transport; needs assessment; geo-spatial; global engineering; Rwanda; Liberia; rural access; GIS





## 1. Introduction

Globally, nearly one billion rural residents lack transportation access to essential services [1]. Without the physical infrastructure to connect households to markets, employment opportunities, health facilities, and educational institutions, households are at risk of experiencing devastating barriers to their economic and social livelihood. A recent United Nations report describes the important role of infrastructure in connecting services, highlighting that networked infrastructure supports 72 percent of the 169 Sustainable Development Goal targets [2]. Though 80 percent of the world's poorest people live in rural areas [3], "last mile" access, which typically consists of pedestrian and motorcycle access in rural areas, is often overlooked in long-term infrastructure planning [4].

Bridges to Prosperity (B2P), a non-profit organization, provides last mile access to rural communities by building trailbridges that can be used by pedestrians, motorcycles, and livestock. B2P has connected more than one million people globally to critical services through the construction of over 350 bridges in 20 countries. A randomized controlled trial highlighted some of the impacts of B2P trailbridges, finding a 75 percent increase in farm profits, 36 percent increase in labor market income and a 60 percent increase in women entering the labor force [5].

As part of the operational strategy in launching a new country program, B2P conducts a needs assessment to determine scope and distribution of trailbridge need. Historically this approach has been entirely field-based, relying on relationships with local officials, who provide locations where bridges are needed, in addition to details and context regarding the transportation challenges faced by communities within their jurisdiction. Two case studies of B2P's needs assessment process completed in Rwanda and Liberia are examined here, including the pilot of a new remote method, using remote geospatial methods to collect and analyze data. The aims of this study were to understand the extent of the need for trailbridges in rural Rwanda and two counties of rural Liberia, and determine if simple remote methods of bridge site identification are viable and comparable to existing field-based methods.

At 50 people per square kilometer, Liberia's population density is average for Sub-Saharan Africa, with about 50 percent of the population living in rural areas [6]. While the terrain is relatively flat, Liberia's long and intense rainy season creates challenges for rural residents in reaching destinations essential to healthcare, education, agriculture, and economic opportunity. Inadequate farm-to-market access exacerbates low agricultural productivity for the 60 percent of the population that rely on agriculture for their livelihood [7]. Adequate access to essential services is critical for the Liberia population, where nearly 41 percent of the population live on less than $1.90 per day [8]. Given the exceptional challenges faced by Liberia's rural communities, in January 2019, B2P conducted a feasibility study and nine-month assessment of trailbridge need, in order to investigate Liberia as a potential future program. Figures 1 and 2 describe the location of Liberia and the specific counties included in the study.

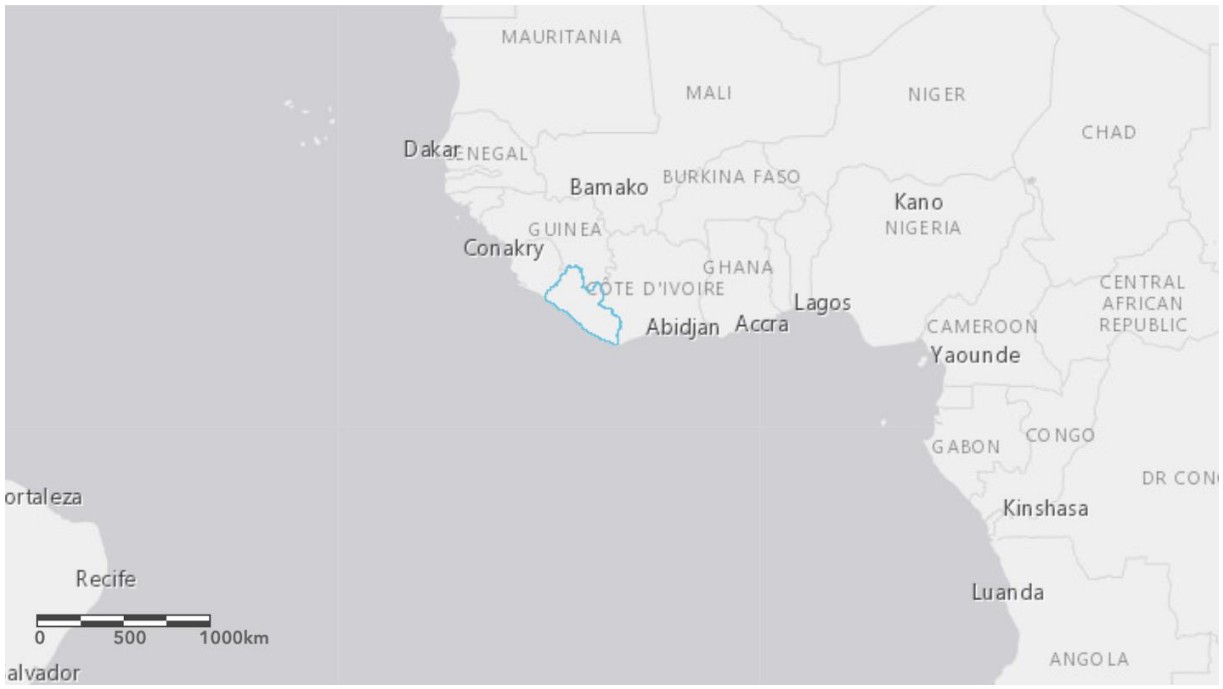

**Figure 1.** Location of Liberia in West Africa. Liberia was one of two countries where needs assessment methods were tested and documented. Prepared by authors using ArcGIS Online [15].

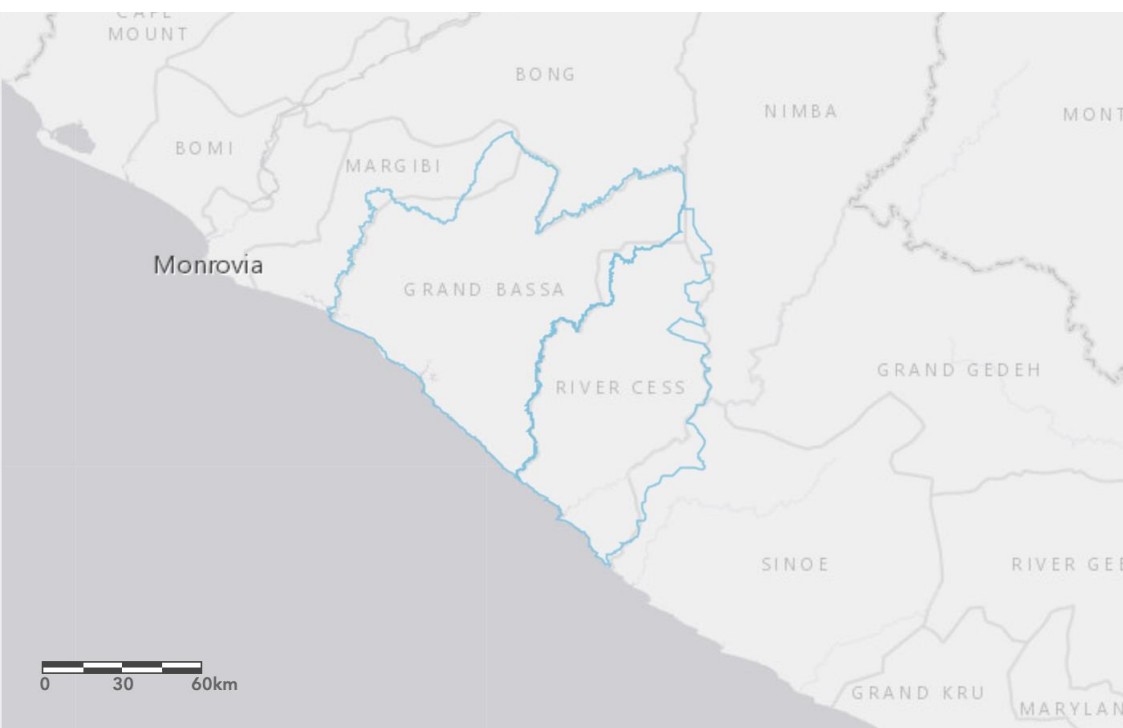

**Figure 2.** Assessment area in Liberia: Grand Bassa County (northwest highlight) and Rivercess County (southeast highlight). Prepared by authors using ArcGIS Online [15].

The overall objective of the Liberia assessment was to test and ground truth new remote bridge identification methods in two counties (Grand Bassa and Rivercess) while simultaneously conducting a needs assessment using a field-based approach previously used in Rwanda. The two counties were selected because they are representative of the country in terms of population density and distribution, and the prevalence of rural farming communities. Additionally, the proximity to the capital city Monrovia meant that field coordination and transportation was possible throughout the period of investigation, and due to the presence of organizations that provide services to rural populations in those counties, some data related to routes and transportation barriers were also available. In addition, the recently completed nation-wide Rwanda assessment provided an additional data set to assess the new remote method.

By contrast, Rwanda is the most densely populated country in Africa (498 people per square kilometers), and the majority (82 percent) of residents live in rural areas, and largely travel by foot [9]. The dense population, coupled with multiple rainy seasons and mountainous terrain creates an environment with an extensive need for pedestrian water crossings. The World Food Programme found that only 4 percent of sampled villages in Rwanda had a market at the village level, and in villages without a market, the average time to reach the nearest market was 86 min [10]. Figure 3 shows the location of Rwanda. While B2P has operated a trailbridge program in Rwanda since 2012, an agreement with the Government of Rwanda to scale its program prompted a nation-wide assessment of trailbridge need in 2018. The objective of the Rwanda needs assessment was to determine the extent and locations of needed trailbridges in Rwanda, which would contribute to a long-term strategy for solving the country's rural isolation problem. While methods for assessing pedestrian transportation access in environments have been extensively documented [11–13], there are significant gaps in the assessment of transportation access in remote and rural environments, particularly for those traveling by foot or motorcycle. The World Bank's Rural Access Index is a valuable metric for estimating rural access in general, but does not provide guidance for identifying specific transportation barriers faced by rural communities. Work conducted by the Overseas Development Institute has demonstrated the relationship between isolation and poverty, and catalogued the myriad dimensions of

isolation that influence a community's ability to access essential services, but incorporates distance and the extent of road networks as important variables, and does not attempt to assess the specific transportation barriers that may be encountered on trail networks [14]. This paper documents a well-tested field-based method for assessing trailbridge need, as well as the first three phases of development and testing for remote methods to assess trailbridge need in the rural context.

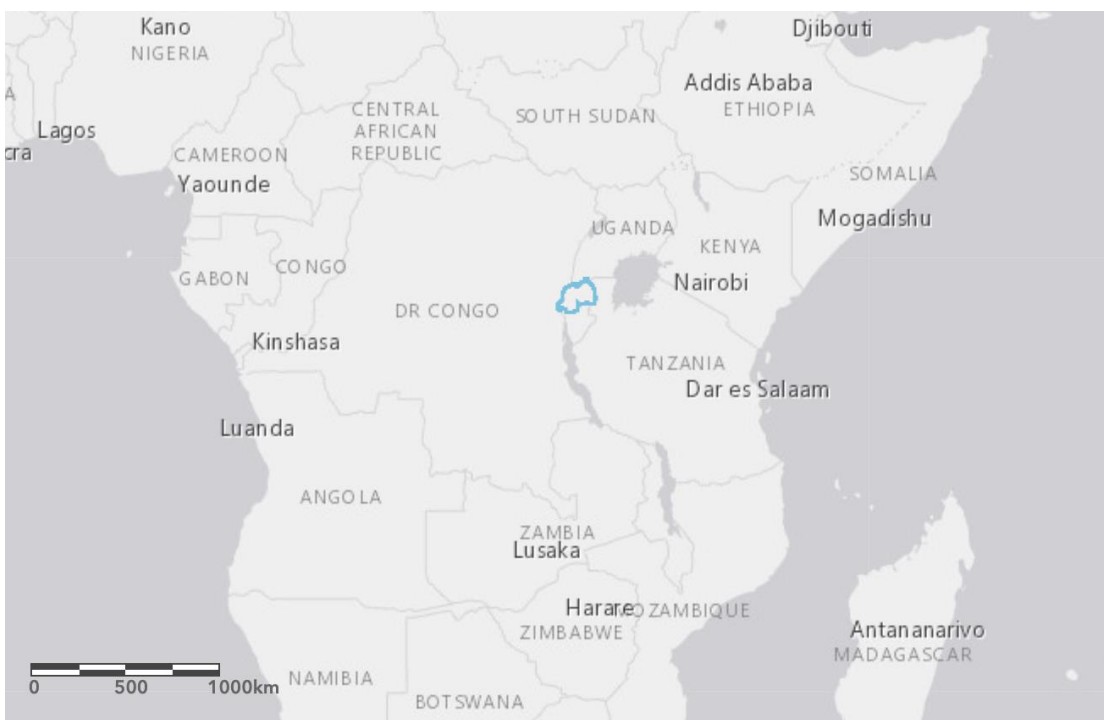

**Figure 3.** Location of Rwanda in East Africa. Rwanda was one of two countries where needs assessment methods were tested and documented. The entire country was assessed. Prepared by authors using ArcGIS online [15].

## 2. Materials and Methods

### 2.1. Field-Based Bridge Site Identification Methods

B2P's field-based method for assessing trailbridge need is adapted slightly for each program country, and considers administrative and governance structures, telecommunications infrastructure, ease of mobility, and social and cultural norms. Ultimately, B2P's goal is to obtain information about potential trailbridge sites from residents and officials who are geographically closest to each crossing and are directly affected by the lack of access. However, given the logistical challenge of visiting each community in a given geography, B2P's process intentionally allows for information to be disseminated to village leaders or chiefs, and bridge requests to be consolidated at each administrative division. The field-based approach has the added advantage of engaging stakeholders at multiple levels in the site identification process, which presumably results in a more complete catalogue of crossing sites. However, this method relies heavily on the local administration's capacity to participate, which can vary significantly across countries, and to an extent, administrative divisions within countries. While effective, this method also presents the possibility for misinformation or incomplete information to be transferred among local officials.

In Rwanda, where the needs assessment took place during the latter half of 2018, an official letter was submitted to each district to introduce the needs assessment and document the support from the national government. Using contact information provided by the district, assessors at B2P's call center would work with leaders at multiple administrative levels of local government to collect information about potential bridge sites, in an effort to vet sites early in the assessment process. Early vetting was conducted primarily

to determine if an identified crossing was not an existing heavy vehicle crossing, and that there was not a serviceable all-season crossing within 300 m.

In Liberia, this process was very similar, however, due to insufficient telecommunications infrastructure in the rural areas of Liberia, requests were collected using pre-mobile phone methods. B2P staff met in-person with leaders at the district level to explain the assessment and provide bridge request forms, which were distributed at the clan and village level. When B2P received the final list, a comparison of the requests received from the clans revealed several areas with low responses. Further investigation found that several clans did not receive instructions to submit bridge requests, and so a direct follow up with each of these clans was conducted to develop a full list. This degree of coordination is typical of field-based assessments in all countries in which B2P has conducted needs assessments. The process for bridge site identification using the traditional method is illustrated in Figure 4. Following the identification of sites via this method, 30 sites from each county were randomly selected for validation and assessment.

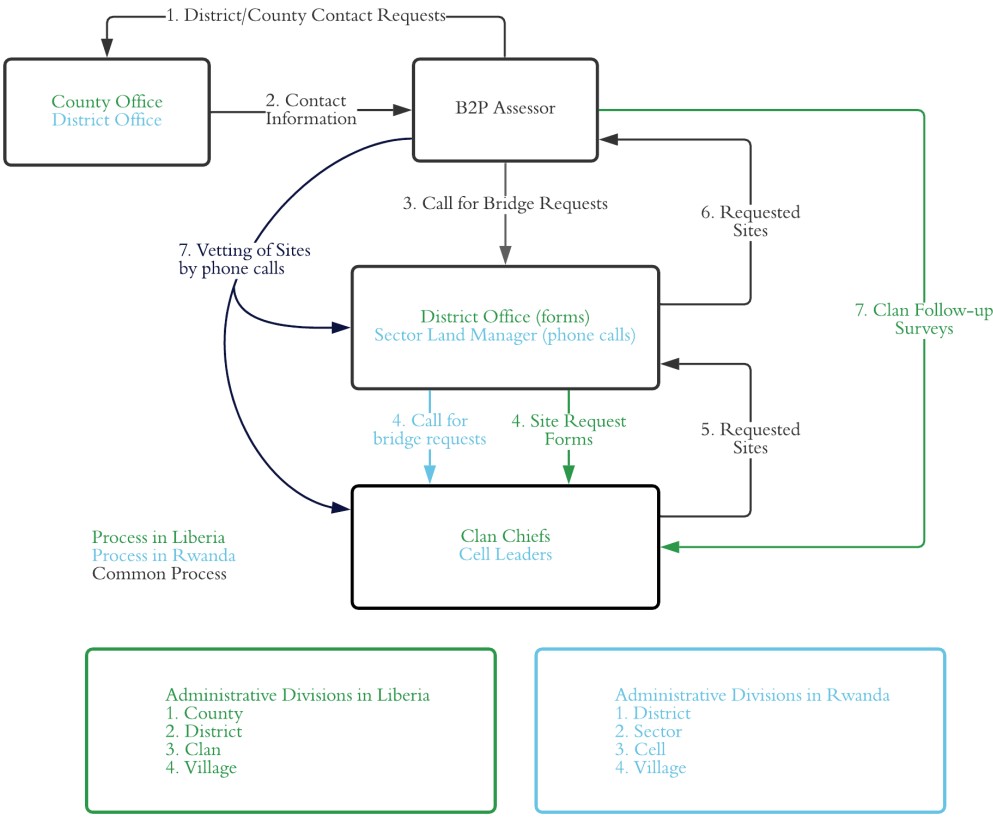

**Figure 4.** Site request process, with relevant administrative divisions in Liberia and Rwanda.

Participation in focus groups was not compulsory, and assessors made it clear that project confirmation was ultimately up to the government and that focus group participation would not influence the likelihood of a bridge being built at the identified site. Personally-identifiable information (including photos, names, or other personal information) information was not collected. All work was conducted in accordance with B2P's programmatic objectives, and in line with established government partnerships.

### 2.2. Remote Bridge Site Identification Methods

While some level of local knowledge will always be critical to identifying bridge need, for B2P, the process is constrained by the time required to coordinate and implement field activities, and as such, limits the ability to produce a nation-wide inventory of trailbridge need. There is also a risk of missing sites that may be important to communities, but unknown to or deemed unimportant by their representatives. In order to reduce

the initial time and resources required to assess trailbridge need in a given geography, and to capture a comprehensive catalogue of sites without the influence of local politics, B2P developed a cost-effective remote method to identify sites where trailbridges may be needed using available spatial data and satellite imagery. This work was developed and extensively tested and iterated in Liberia, and the most promising of the three trial methods was later tested on a limited area of interest in Rwanda, where B2P already had a comprehensive catalogue of identified sites.

Using freely-available data and geospatial analysis techniques, the objective of the remote bridge site assessment was to systematically identify and catalogue the locations of needed trailbridges, initially in the assessment area of Grand Bassa and Rivercess Counties in Liberia. It is worth noting that spatial data in Liberia is limited in both general availability and quality; key datasets such as route extent, classification and condition, and waterways, are largely incomplete and unreliable. This made Liberia an ideal proving ground for remote site identification, as it represents an extreme end of the spectrum in terms of data availability, and as such, forced new approaches that may be useful in other geographies. Three methods developed using QGIS were evaluated for remote site identification [16].

### 2.2.1. Method 1: Prediction Using Zonal Statistics

The zonal statistics approach, which compared raster values (data represented in matrix of cells in which each cell represents a value or attribute) in a defined area, uses support-vector machine (SVM) aggregated statistics, from the regions surrounding previously completed training trailbridge sites, in order to predict the locations of sites where waterways impede pedestrian access. The training trailbridge sites, and the corresponding statistics, were drawn from B2P's global database of completed trailbridges. In total, training data included GPS coordinates and other metadata from 291 trailbridge sites in 21 countries. Each site's point location was buffered with a two-kilometer radius, and then statistics were calculated within the resulting polygon. The statistics calculated for each completed bridge include estimated population [17], land cover classification [18], and river characteristics [19]. Figure 5 illustrates the two-kilometer radius created around a single site, and each discrete layer of data for which statistics were calculated within the radius. Table 1 describes the aggregated statistics from the site shown in Figure 5.

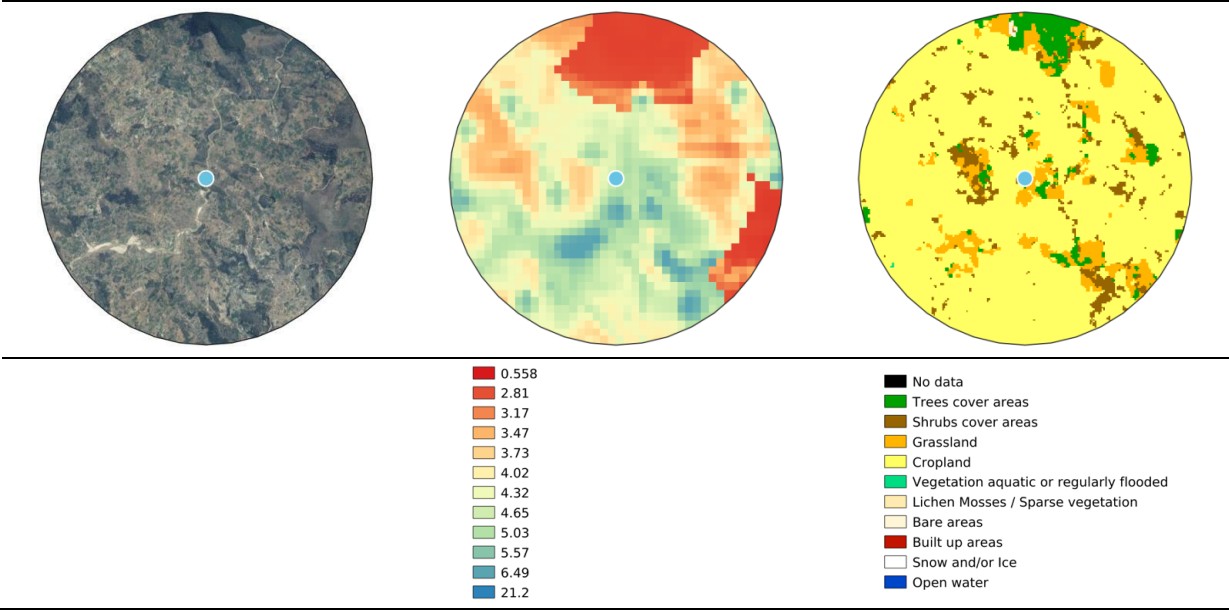

**Figure 5.** Two-kilometer radius around training Nkuri bridge site in Rwanda, with satellite imagery, population visualized as people per pixel, and land cover, left to right. Prepared by authors using QGIS [16].

**Table 1.** Aggregated statistics for number of individuals and land cover within the two-kilometer radius around training Nkuri bridge site, illustrated in Figure 5.

| Bridge Site | Nkuri, Rwanda |
|---|---|
| Population | 6021 * |
| % Cropland | 84 ** |
| % Forest | 5 ** |
| % Built Up | 2 ** |
| % Other | 9 ** |

* WorldPop. ** European Space Agency (ESA).

Based on the characteristics identified using zonal statistics, a support-vector machine (SVM) model was built to predict whether a specific location was a potential trailbridge site. The model used data collected as part of B2P's needs assessment in Rwanda, as training data for valid and invalid bridge sites. Once trained, the model was evaluated at a 2 km interval along Liberian waterways using the same zonal statistics datasets as Rwanda, and gave a predicted binary output for each point.

### 2.2.2. Method 2: Prediction Using Route-Based Accessibility Method

The route-based accessibility method identifies locations where a waterway impedes the access of a population to essential services by routing population clusters to the nearest destinations for accessing healthcare, education, farms, or towns where markets or wage labor opportunities are located. Routing was conducted on an existing plotted route network, and a relative accessibility score was calculated for each population cluster. This approach required the aggregation of externally-sourced data sets, including route networks (both roads and footpaths, from government, NGO, and open sources), waterways, and locations of communities, towns, hospitals, and schools in order to compute the routes of populations from where they reside to the nearest services. Figure 6 illustrates the route-based accessibility method.

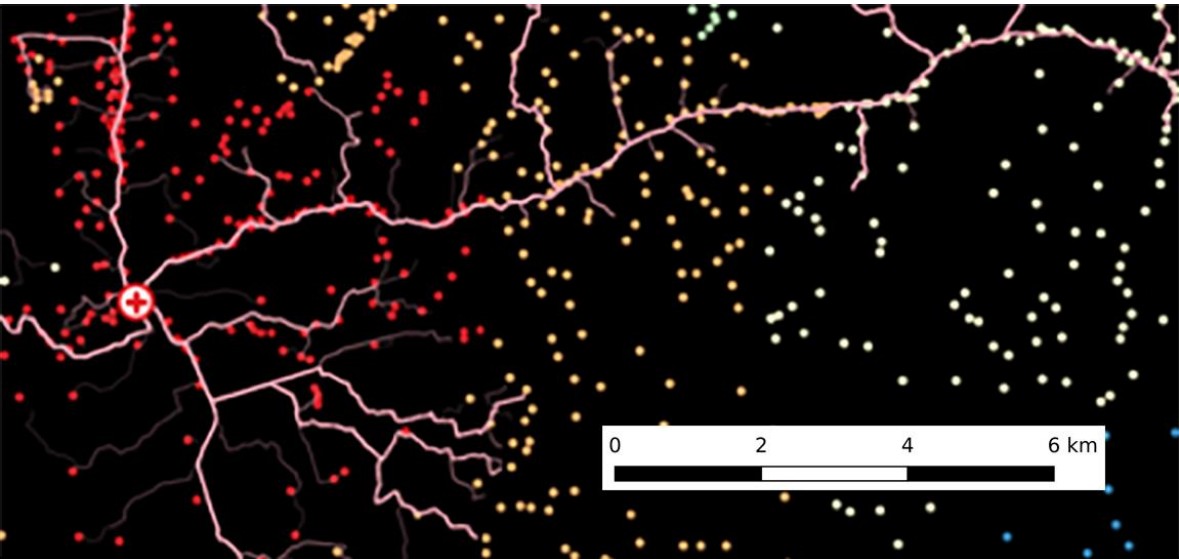

**Figure 6.** Routes to the CH Rennie Hospital in Kakata, Liberia from nearby communities, colored cool to warm according to their distance from the hospital. Prepared by authors using QGIS [16].

A closest-distance route from each of the communities in the data was calculated using route network data to determine the nearest hospital, school, and towns, and the distance traveled to each service was calculated using a route network. This set of community distances to each service was then min-max scaled and coalesced into a [0, 1] community accessibility score. The accessibility scores reflected how accessible each community was

to its nearest essential services and were used as control points to build a continuous accessibility surface for an entire region. Finally, the waterways dataset for the region was traversed, and at every two-kilometer interval, the accessibility surface on either side of the waterway was sampled 250 m from the waterway on either side. The locations with the highest difference of accessibility between one side of the river and the other were identified as the potential bridge sites.

### 2.2.3. Method 3: Machine-Assisted Site Identification

Methods 1 and 2 relied on freely available data sources from government agencies, NGOs, and providers such as OpenStreetMap; this data enabled insight into a region in which B2P had not yet conducted field work. However, the main difficulty with the geospatial approaches stemmed from data quality and availability. While key datasets are available, the quality varies widely from region to region and often has the poorest quality and density in the remote areas where B2P focuses its work. The dearth of spatial data in these regions makes automated prediction with any analysis approach difficult. It was determined that in the absence of high-quality data, there would be a need for some level of manual intervention. In service of this, B2P developed a machine-assisted method, which merged automation and human intervention for remote site identification.

Method 3 involved the division of the assessment area into grid cells, which were then systematically investigated using satellite imagery overlaid with ancillary datasets to aid in bridge site identification. Figure 7 illustrates the machine-assisted site identification method.

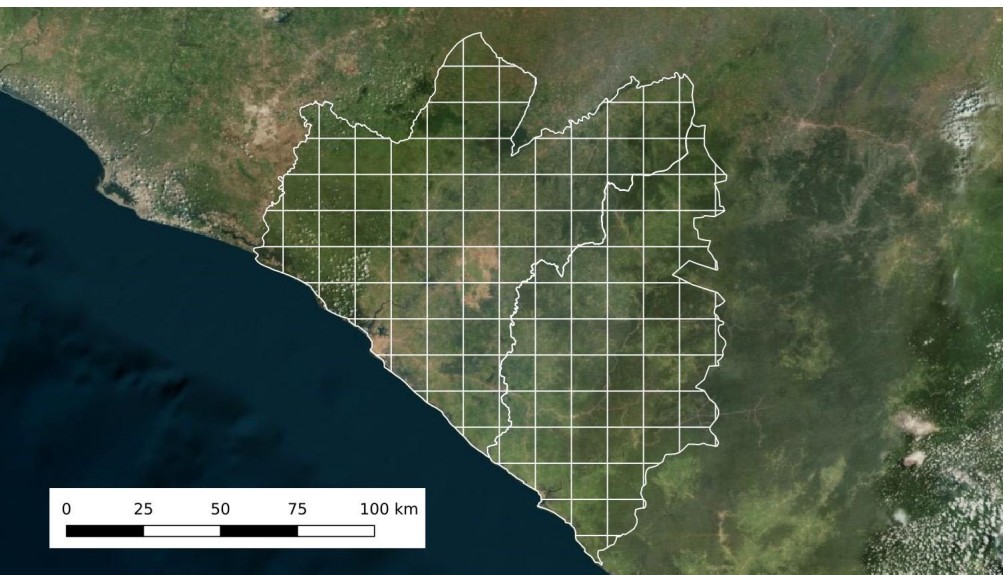

**Figure 7.** Grand Bassa and Rivercess counties divided into 0.1 degree grid. Prepared by authors using QGIS [16].

In the exploration of the Grand Bassa and Rivercess counties in Liberia, Bing satellite imagery captured in 2013 was overlaid with multiple routes, waterways, and communities databases, as well as the route-based accessibility surface to create an exploratory environment from which to investigate bridge need. Figure 8 shows an example of a singular crossing identified using the machine-assisted site identification method.

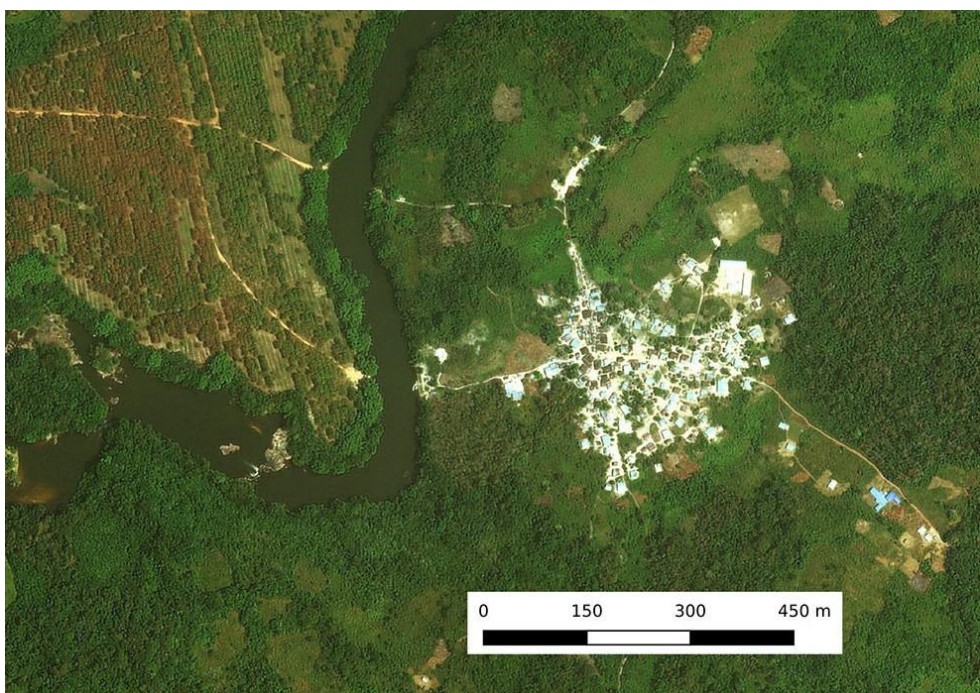

**Figure 8.** Example of a crossing at Jacob Town, Grand Bassa identified by Method 3. Prepared by authors using QGIS [16] a.

### 2.3. Validation Procedure for Remotely Identified Sites

Sites identified through both remote methods and by government officials were randomized for validation and site assessment. The sites predicted using remote methods were randomized by prediction method and county, and a subset of those randomized sites was assigned to the field team for a site assessment. To validate a site, a field assessor traveled to the location of the coordinates provided (or as close as safely possible), and using a digital form on a tablet, recorded whether a potential trailbridge site existed at or near that location. In this case, a potential trailbridge site was defined as any location where a route used by pedestrians crossed a waterway. Sites identified by government officials were considered valid if the coordinates or location returned by the identification method was at or very near a water crossing, regardless of whether there was an existing all-weather crossing. If the community or local officials perceived a bridge at the crossing as unnecessary, but the waterway was not crossable year-round, it was still considered valid and was included in the summary of potential sites.

When a site was deemed to be valid, it was further assessed to determine if an all-weather crossing was required, and if so, the recommended solution based on the results of the technical assessment. A site was defined as having an existing all-weather crossing if there was a bridge in place (trail or vehicular), and if local residents confirmed that the bridge was safely passable year-round, regardless of the flood-level of the river. If community members or local officials perceived a bridge at the crossing as unnecessary, but also noted that the crossing was sometimes impassable or required fording water, it was included in sites that require an all-weather crossing. Examples include existing bridges that were overtopped during periods of heavy flooding, or rivers that flooded infrequently. This was done to ensure consistent application of the criteria to determine a need for safe crossing, independent of local norms and risk tolerance for travel.

### 2.4. Site Assessments

If a site was deemed valid by either method, field team members conducted a full site assessment, which consisted of a technical and social component. The technical component included choosing a proposed centerline for a potential trailbridge, determining

the highest point that the water level had ever risen (as indicated by elder local residents) to determine the bridge design type and estimated span, and taking photos along the proposed centerline of the bridge, to aid more practiced engineering staff in evaluating initial technical determinations. The measurements taken in the field by assessors were input into a spreadsheet tool that can provide a rough preliminary bridge design and confirm the type of bridge necessary.

The social assessment followed a focus group protocol in which assessors gathered information by sketching a map of the area with the group and marking the crossing location; communities that use the river crossing on a weekly basis; important destinations; and walking paths or roads. Community representatives at the meeting provided the populations of the communities they represented, and approximate walking times to the crossing point. Information about other communities not represented at the meeting was estimated by those present. Additional information was collected, including seasonal accessibility, primary occupations, and crops grown or animals raised by community residents or nearby large farms. Figure 9 shows an example of a community map created during the focus group sessions.

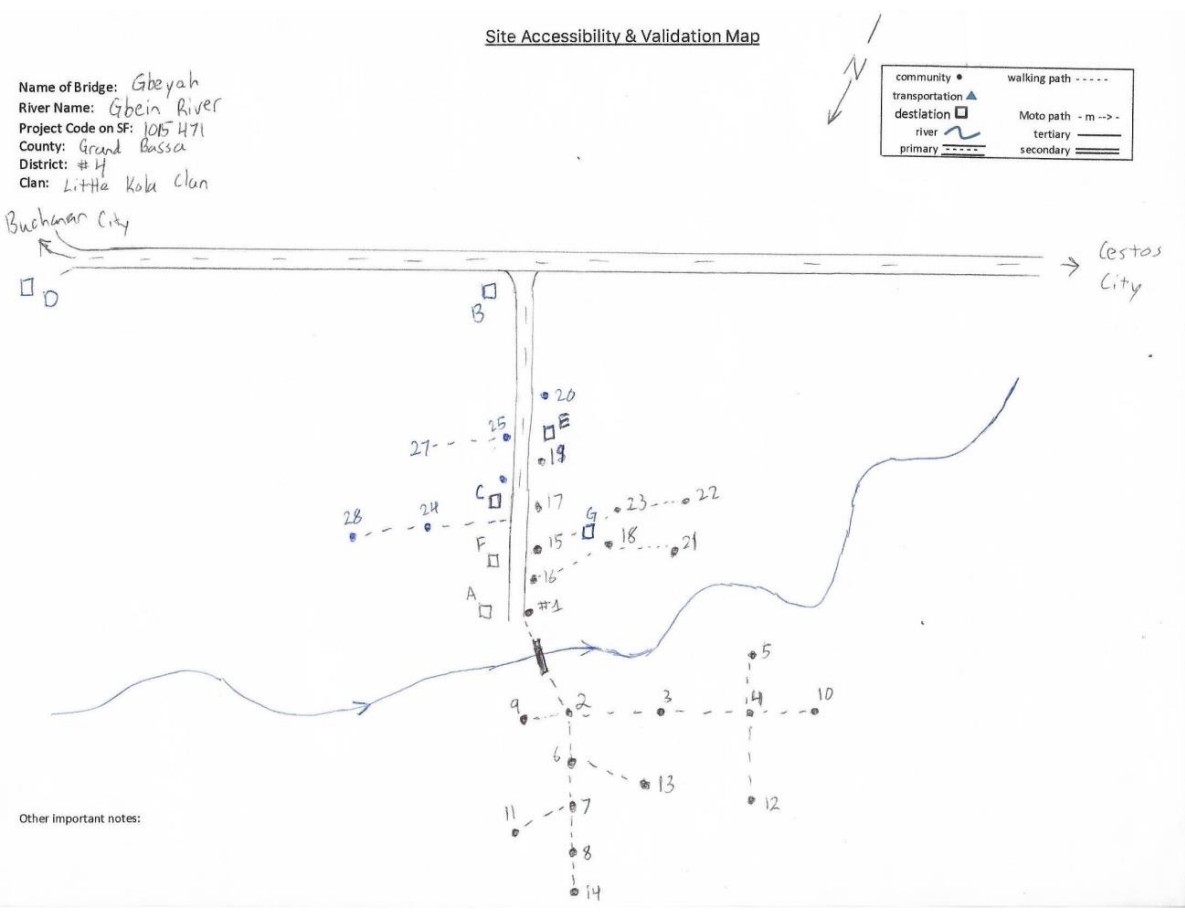

**Figure 9.** Example of a map created during a community focus group; maps were created by B2P field assessors during focus groups. Numbers at points represent communities or destinations, which were logged and numbered, and in some cases, visited and documented.

The technical and social assessment methods used in Liberia in 2019 largely followed the same protocols used in Rwanda in 2018.

In Liberia, B2P conducted a more in-depth network analysis, to gain an in-depth understanding of the destinations deemed important by community members, and the distance of those key destinations from the reporting communities. This work also informed the design of B2P's population catchment study conducted in Rwanda later in 2019 (full re-

port forthcoming). Following the completion of the standard social assessment, the field team collected a comprehensive set of data related to destinations deemed important by communities during focus groups. In addition to recording comments about important destinations as part of the social assessment, assessors visited each destination that was feasible and safe to visit, and recorded a geolocation, a photo, the category and type of destination and any other known characteristics, such as the number of students enrolled in a school, or the types of services offered at a health facility. Destinations included schools, health centers, markets, charging stations, transportation junctions, government offices, religious buildings, water sources, places of employment, and various other destinations deemed important by members of the focus group.

Extremely remote destinations, for which travel would have been particularly time consuming or potentially dangerous, were not visited, and in those cases the geolocation was identified using satellite imagery, as well as with data provided by locally operating organizations. Assessors recorded the geolocation of a community if they were already in close proximity, but due to the large number of remote communities that were accessible only by walking several hours, recording coordinates for most communities was not feasible. B2P accounted for this by comparing the data in existing communities' datasets to the communities data collected by assessors as part of the focus group process.

*2.5. Data Audit*

Generally, in both Liberia and Rwanda, social data was found to be consistent between the original and duplicated assessments, with similar estimations of the number of crossing-related deaths and injuries, and described characteristics of the communities (in terms of population, occupations, crops grown, and important destinations). In Rwanda, some inconsistencies related to the duration and number of flood events were observed, though those inconsistencies were corrected with the rephrasing of the initial question and this modification was carried over to the Liberia assessments.

The data audit revealed the technical component of the assessment to be more challenging for the assessors. B2P's technical assessment method requires recently hired assessors to determine the optimal placement for bridge abutments at a given site and take vertical distance measurements with a laser range finder, which measures to an accuracy of 0.2 m. This preliminary information makes it possible to determine the feasibility of trailbridge construction at the crossing and establish the span and bridge type in order to estimate the cost of construction. Determining the placement of the abutments, though simple for practiced staff, requires a significant amount of field training, and presents a significant challenge for new assessors, and by proxy, for B2P's needs assessment work overall. By contrast, a final bridge design is created with computer-aided design software, using centerline profiles created with data collected as part of a full technical survey, which takes significantly more time than a standard technical assessment and is not feasible for the needs assessment phase. (Full surveys are typically conducted once B2P is prospecting a site for construction in the immediate future.)

Because the initial technical assessment, though effective, requires long training periods at a significant number of sites (particularly for mountainous regions where steep slopes make abutment placement more challenging), B2P is exploring the option of using new technologies such as drones, lidar, and photogrammetry to develop 3D models that would allow more experienced engineers to complete preliminary designs remotely, or produce designs automatically.

**3. Results**

*3.1. Field-Based Bridge Site Identification Results*

In Rwanda, the total number of sites assessed by Bridges to Prosperity during the 2018 nation-wide assessment was 1447 sites. In total, 1518 bridge requests have been validated in Rwanda (including sites previously completed or identified by B2P, and not included in the most recent assessment), which amounts to approximately 1 trailbridge needed for

every 17 km² of land. The vast majority, approximately 80 percent, of identified water crossings in Rwanda are over narrow streams that experience flash flooding and repeatedly wash away local timber bridges but are not large enough waterways to require the type of cable suspension bridges B2P typically builds. Of the 1447 sites assessed, 20% required a cable suspension design while the remaining 80 percent were better suited for simple short span bridges of 10–20 m. A typical crossing in Rwanda where a bridge was requested is shown in Figure 10.

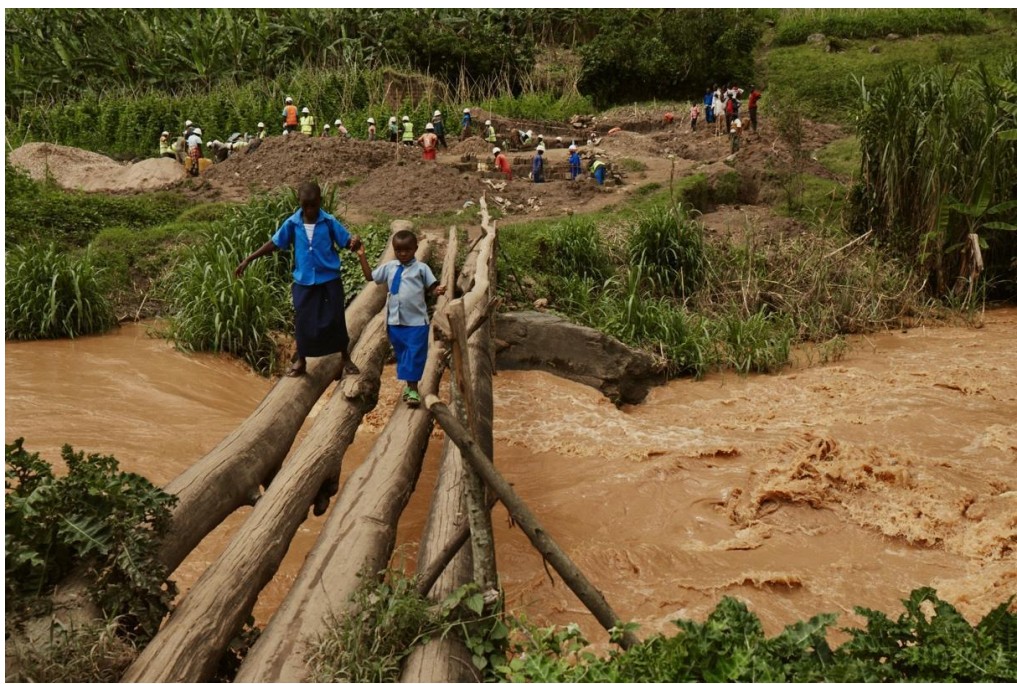

**Figure 10.** Typical local crossing in Rwanda, with new bridge construction in background. In Rwanda, it is common for communities to build to timber bridges like the one picture. Simple timber bridges are frequently overtopped or washed away during periods of flooding. Photo by Collin Hughes.

In Liberia, a total of 1062 sites were identified in the two counties of focus, through using the field-based method, which relies on local government. Figure 11 shows a typical crossing in Liberia where a bridge was requested. Of those, 83 percent were located in Grand Bassa County and 17 percent in Rivercess County. Of these, 30 sites from each county were randomly selected for validation assessment, as described Methods. In total, only one of the government-identified sites randomized for field assessment was determined to be invalid (meaning that the assessor could find no crossing at the described site), though some crossings had a vehicular bridge present, which meant that a trailbridge was no longer needed. Similar to Rwanda, though to a lesser extent, a large portion of sites (48 percent) were located at small streams, where a cable suspension bridge would not be an efficient bridge design. Cable-suspension bridges would be required at 38 percent of identified sites. In Liberia, all site validation was conducted in person due to limited telecommunications infrastructure. Of the sites identified via the field-based method, 13 percent of resulted in a location where an all-weather crossing (usually a vehicular bridge) had been constructed or was invalid, meaning there was no barrier along a road or walking path. Of the randomly selected sites, 82 percent needed a trailbridge ($p < 0.001$), and extrapolated to the full area of the two counties, we estimate a total of 867 trailbridge sites are needed in total, or approximately one trailbridge needed per 12.7 square kilometers. An example trailbridge which could be constructed to replace the typical timber crossings in Rwanda and Liberia is shown in Figure 12.

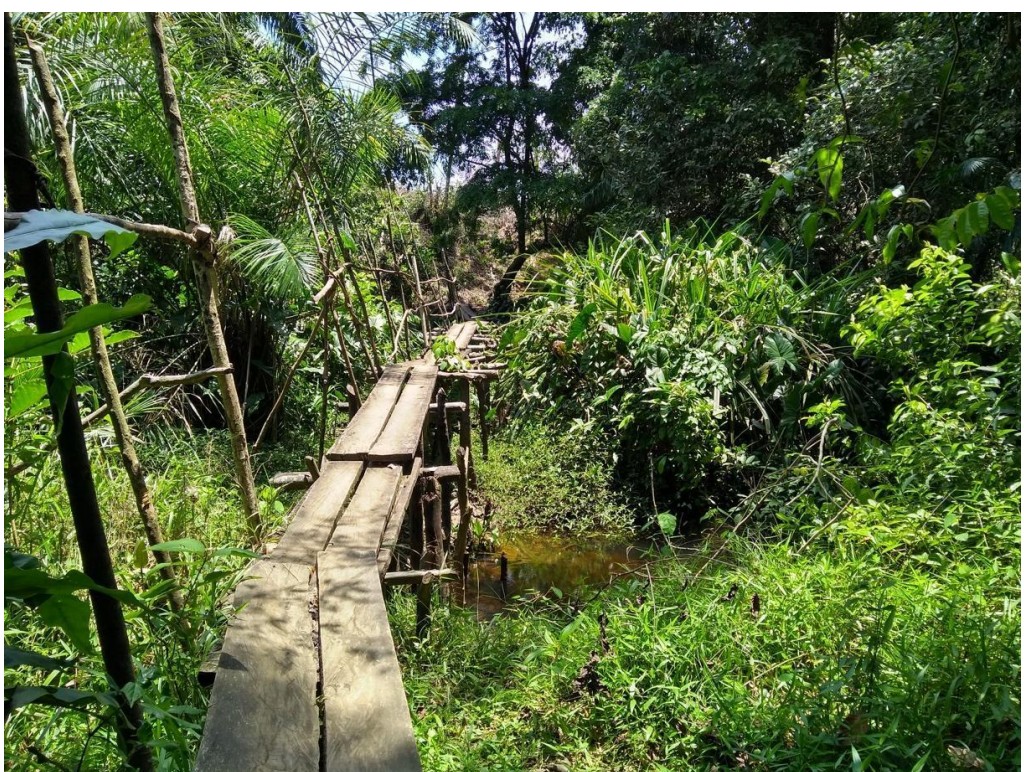

**Figure 11.** Typical local crossing in Liberia. While effective during the dry season, crossings built by communities are frequently overtopped or washed away during periods of flooding. Photo by authors.

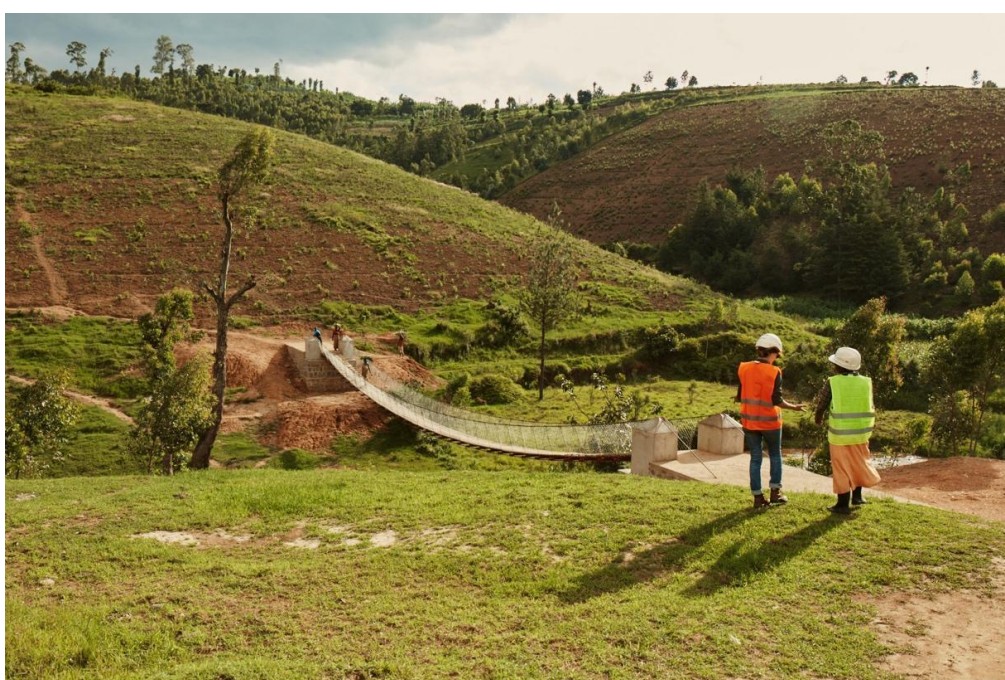

**Figure 12.** B2P suspended trailbridge in Rwanda, one of two standard cable suspension designs. B2P trailbridges are designed according to the historic high water level, derived from bank characteristics and community reports. Photo by Collin Hughes.

*3.2. Remote Bridge Site Identification Results*

3.2.1. Method 1 Results: Prediction Using Zonal Statistics

The sites identified by zonal statistics prediction were manually viewed on satellite imagery to determine if there was a visible river and pedestrian crossing. In this manual review, it was clear that the model was not effective at identifying bridge sites. Overall, only 12 percent of the sites predicted by the model were determined to be sites where pedestrian water crossings were located.

Upon further investigation, it appeared that the model was predicting the location of a bridge site based upon population, as it is likely that the model used land cover distribution as a proxy for population in addition to the population dataset. Through this exercise, it became clear that applying zonal statistics is too narrow of a context for bridge site prediction. Bridges provide connection from one place to another, so a predictive model must consider how a potential site fits into the overall route network, as opposed to viewing a river crossing as a discrete point, without the additional context of communities, destinations, and the routes between them.

3.2.2. Method 2 Results: Prediction Using Route-Based Accessibility Method

While this method had promising results in other regions where it was tested to a more limited degree (including Lesotho, which has a higher availability of spatial data that rendered accessibility gradients that were able to predict the location of viable trailbridge sites), the automatic gradient detection failed to produce viable results in this analysis. (The validation of results was conducted manually, using satellite imagery.) As this method was reliant on high-quality and complete vector data, it was not able to render an accurate prediction using the low-quality data available for Liberia.

One telling indicator of route data availability is the average distance between each community and the nearest road: in Grand Bassa, the average snap distance (the distance between two points, along the nearest mapped routes) is 3.1 km, whereas it is 1.7 km in Rwanda and 0.2 km in Lesotho. Due to this large snap distance, locations of steep accessibility gradients are located far away from where the obstacles are actually located and are unable to be detected by the automatic waterway traverse method. Though this method was unable to predict the trailbridge locations entirely automatically, the accessibility surface generated by this method still proved useful when incorporated into the machine-assisted bridge site identification method. In the future, this method may be more useful if regions where a higher volume and quality of spatial data is available.

3.2.3. Method 3 Results: Machine-Assisted Site Identification

The results from this method identified a total of 70 sites throughout Grand Bassa and Rivercess counties in Liberia. Given the manual component of this method, it was useful to also track the number of hours to complete this analysis (27 h) and an average area processed per hour (501 square kilometers). The results were promising, as they were achieved relatively quickly (despite the manual effort required), and the output was a list of precise locations, rather than predication areas. Sparse population distribution meant that there were few major routes and pre-existing river crossings, with only major ones visible by satellite, meaning that in this context, we believe it would be difficult for automated methods to identify locations with the same accuracy and precision as a human. The limitation of this approach is that motorcycle and foot paths are often obfuscated by dense vegetation, and the routes and their waterway crossings are not visible by satellite. While the algorithmic prediction of bridge need is a promising endeavor, there are simply too many factors that influence bridge need that are not reflected in the available data. However, these results showed promise in using satellite imagery assisted by algorithmic outputs in order to identify potential trailbridge need. Given that the machine-assisted method was the only method deemed successful, all 69 sites identified during the machine-assisted method process were used in identification and validation of bridge sites. Of the

69 sites identified by Remote Method 3, 15 sites in each county were randomly selected for validation by assessors. No remotely identified sites were deemed invalid ($p < 0.001$).

Following the Liberia assessment, Remote Method 3 was conducted in Nyamasheke and Rusizi districts in Rwanda, which identified 1183 sites. By comparison, B2P's 2018 needs assessment in Rwanda identified 185 sites in the same districts. Though no field validation was conducted following the remote assessment in Rwanda, 100 percent of the 300 sites randomly selected for a second manual check using satellite imagery appeared to be valid crossings. Additionally, 72 percent of sites that had been identified as part of the 2018 Rwanda needs assessment were identified by Remote Method 3, as well as 100 percent of sites that were determined to be appropriate for B2P's standard cable suspension bridge designs.

### 3.3. Validation Results for All Identification Methods

Beyond initial site validity, which only considered whether a crossing was actually present, all identified and assessed sites in Liberia were further analyzed to determine if an all-weather crossing was actually needed based on the stated criteria and the presence of any existing infrastructure, and if so, what trailbridge design would be most appropriate. Of the valid sites, 24 percent had an existing all-weather crossing (in this case, vehicular bridges that the community reported as not being overtopped by the river during floods), and 76 percent were in need of an all-weather crossing. Two randomized sites that were confirmed by community members to be valid crossings (in that they were locations where pedestrians crossed a waterway along a route) were not visited by the assessors, as the sites were prohibitively far away. As such, those sites were excluded from the analysis to determine which of the identified sites required an all-weather crossing.

Of the sites in need of an all-weather crossing, 44 percent were deemed appropriate for a B2P standard trailbridge design, while the remaining 56 percent would benefit from a short span bridge, similar to the results of the Rwanda field assessment. Both remote and field-based methods had a high success rate for identifying valid water crossings, though the field-based method rendered many more sites. Half of sites identified by the remote method were at locations where a vehicular bridge was already present, highlighting the limitation of using satellite imagery or spatial data to identify crossings. Tables 2–4 describe results of field-based and remote methods in Liberia and Rwanda.

**Table 2.** Comparison of results between field-based method in Liberia and Remote Method 3 in Liberia, both conducted in Grand Bassa and Rivercess Counties.

| | Liberia Field-Based Method | Liberia Remote Method |
|---|---|---|
| Total sites identified | 1062 | 70 |
| Valid pedestrian crossings | 98% ($p < 0.001$) | 100% ($p < 0.001$) |
| Existing all-weather bridge present | 12% | 50% |
| B2P standard trailbridge design inefficient (i.e., short span bridge more appropriate) | 40% | 0% |
| B2P standard trailbridge design insufficient (i.e., estimated span exceeds 150 m) | 2% | 10% |
| B2P standard trailbridge design appropriate | 45% | 40% |

**Table 3.** Results of field-based method in Rwanda, conducted nationwide. Note: Valid pedestrian crossings were likely at 100% due to the vetting process conducted prior to each field visit.

|  | **Rwanda Field-Based Method** |
|---|---|
| Total sites identified | 1447 |
| Valid pedestrian crossings | 100% ($p < 0.001$) |
| Existing all-weather bridge present | 3% |
| B2P standard trailbridge design inefficient (i.e., short span more appropriate) | 74% |
| B2P standard trailbridge design insufficient (i.e., estimated span exceeds 150 m) | 6% |
| B2P standard trailbridge design appropriate | 21% |

**Table 4.** Comparison of results of the field-based method in Rwanda, limited to results from the Nyamasheke and Rusizi districts, to Remote Method 3 in Rwanda, which was only conducted in those two districts.

|  | **Rwanda Field-Based Method** | **Rwanda Remote Method** |
|---|---|---|
| Total sites identified | 185 | 1183 |
| Valid pedestrian crossings | 100% ($p < 0.001$) | 100% ($p < 0.001$) |
| Existing all-weather bridge present | 0% | N/A |
| B2P standard trailbridge design inefficient (i.e., short span more appropriate) | 78% | N/A |
| B2P standard trailbridge design insufficient (i.e., estimated span exceeds 150 m) | 0% | N/A |
| B2P standard trailbridge design appropriate | 22% | N/A |

## 4. Discussion

### 4.1. Efficacy of Field-Based Methods Versus Remote Methods

In Liberia, though the validity of sites identified via field-based methods or remote methods was similar, the field-based method produced a much larger catalogue of sites, and more accurately represents the scope of need, as the remote method failed to detect over a thousand valid sites in the same geography. While Remote Method 3 was ultimately effective in identifying valid water crossings, it identified only 16 percent of valid crossings identified by the field-based method, indicating that any application would be useful in addition to, rather than in lieu of, the described field-based method. Though the results were more promising in Rwanda, where Remote Method 3 identified 72 percent of sites that had been identified as part of the field-based method, and 100 percent of sites determined suitable for B2P's standard design, the risk that a significant number of sites may go unidentified presents too much of an obstruction to be relied on as a sole source of data, even in early phases of assessment. That the remote method did not perform as well in Liberia, where quality spatial data is limited and transportation features in satellite imagery are more difficult for the human eye to detect, suggests that it is limited in regions with significant tree cover.

The field-based method, while costly in terms of time and resources, has the added advantage of incorporating relevant local knowledge and context as well as technical data important for cost estimates, and the opportunity to conduct parallel assessments in the logistical and political conditions that would inform the development of a trailbridge construction program. That said, neither method alone fully encapsulates bridge need and thus the state of the problem necessitates a blended approach with a more sophisticated remote method that is effective in data-constrained environments, and streamlined field-based methods that incorporate existing local knowledge and expertise. In general,

this work builds on the usefulness of the Rural Access Index, a measure developed by the World Bank to evaluate transportation mobility, by evaluating the prevalence of a specific transportation barrier, and deploying rapidly developing technologies to generate results more efficiently than traditional methods [1,20]. This work is limited in that it does not describe transportation barriers beyond rivers that may exist between communities and essential destinations, or how road or trail conditions may impede rural access.

### 4.2. Trailbridge Design Insights

Collectively, more than half of sites identified in Liberia and Rwanda were not suitable for B2P's standard bridge designs. The Rwanda needs assessment of 2018 highlighted a significant need for shorter, simpler structures, and served as the catalyst for B2P to consider additional standard designs, and local construction and procurement frameworks. The results of both assessments emphasized the importance of determining the optimal bridge design earlier in the process and point to the need for investing in more sophisticated technical assessment methods in addition to more sophisticated remote site identification techniques.

### 4.3. Application to Rural Infrastructure Development Programs

The high demand for trailbridges in both countries underscores the need for a better understanding of the barriers that rural communities face in reaching critical destinations and opportunities, and by proxy, the need for cost-effective, scalable methods to identify barriers and locally appropriate solutions. Within the transportation sector, there is a dearth of literature and practical resources related to the identification of rural transportation infrastructure, or barriers faced by last mile communities in reaching essential destinations. As such, the authors hope that the methods documented here contribute meaningfully to a body of knowledge that will inform the work of other entities invested in safe and consistent transportation access for rural communities. The identification of transportation barriers is critical to both the development and prioritization of infrastructure projects, as well as the broader understanding of how those barriers affect target communities' access to existing services or interventions.

## 5. Conclusions

This work underscores the need for trailbridges in rural regions, and the degree to trailbridges and other rural transportation infrastructure are both under-resourced and poorly understood, in addition to cataloguing the need for trailbridges in Rwanda and a portion of Liberia, and illuminating well-tested field methods for rural access assessments. Future work in needs assessments should be focused on three outcomes: developing more effective methods for remote site identification; developing more effective methods for determining optimal bridge design early in the assessment process; and incorporating local knowledge and involvement earlier, and in a more streamlined manner. Already, B2P has engaged remote sensing scientists in remote identification efforts that include machine-learning method using neural networks, with promising early results, and industry experts in deploying new technologies that use existing digital elevation models to yield data important to bridge design and cost estimates. In parallel, B2P is working to identify how to best engage government partners in early needs assessment work, to leverage existing expertise and growing resources in data in and analytics, without placing undue burden on entities already facing massive development challenges with limited resources.

**Author Contributions:** Conceptualization, K.S., D.L., A.N., and C.B.; methodology, K.S., D.L., A.N., and C.B.; software, D.L. and A.N.; validation, K.S.; formal analysis, D.L. and A.N.; investigation, K.S., D.L., A.N., and C.B.; resources, A.N. and C.B.; data curation, A.N.; writing—original draft preparation, A.N. and C.B.; writing—review and editing, K.S., A.N., and C.B.; visualization, A.N.; supervision, A.N. and C.B.; project administration, A.N. and C.B.; funding acquisition, A.N. and C.B. All authors have read and agreed to the published version of the manuscript.

**Funding:** This research was funded by The Vitol Foundation.

**Institutional Review Board Statement:** Not applicable.

**Informed Consent Statement:** Not applicable.

**Data Availability Statement:** Data available on request. The data is this study are available on request from the corresponding author.

**Acknowledgments:** The authors would like to thank the governments of Liberia and Rwanda for their support and cooperation; the assessment teams that conducted field work in Liberia and Rwanda; staff at Last Mile Health in Liberia for routes data and key logistical information; Alissa Davis for her editing and narrative support; Joe McGlinchy at University of Colorado Boulder's Earth Lab for his thoughtful comments; and the anonymous reviewers and editor for their careful reading and thoughtful comments on an earlier manuscript.

**Conflicts of Interest:** The authors were all employed by Bridges to Prosperity at the time of the research. The funders had no role in the design of the study; in the collection, analyses, or interpretation of data; in the writing of the manuscript, or in the decision to publish the results.

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
