# Peer review of "Identifying Water Crossings in Rural Liberia and Rwanda Using Remote and Field-Based Methods"

_sustainability, doi:10.3390/su13020527_

Round 1
Reviewer 1 Report
Sustainability REV
This is a very interesting and useful manuscript about the identification of water crossings useful to human populations inhabiting rural areas of Liberia and Rwanda. This type of papers are ethically and technically very important and I think that should be rapidly published to promote strategies of sustainable develpments in these Countries. Ms is well written, with a large amount of data obtained from high research field- and remote sensing efforts. I am not an English Mother Tongue but the ms is readable and I think that the English style and language is good. I have only minor comments to suggest, reported below (I propose MINOR REVISIONS). However I think that, after these changes, the ms deserves to be published on the Journal.
MINOR COMMENTS
- 85-86. Here the authors introduced the problem solving as concept. I think that they should refer to some seminal references about the approaches and techniques in this regard. For example, about conservation and sustainability, see ‘ Toward a new generation of effective problem solvers and project-oriented applied ecologists. Web Ecology, 20(1), 11-17, 2020 and ‘ Unifying the trans-disciplinary arsenal of project management tools in a single logical framework: Further suggestion for IUCN project cycle development. Journal for Nature Conservation, 41, 63-72.’ More particularly in this last paper it has been defined the different steps of a strategy (context, planning, process, monitoring). I think that these steps should be better defined in your paper. For example, Fig 1 refers to a classical ‘situation analysis’ of a project. However a large part of the paper can be considered an interesting ‘context analysis’ aimed to develop a strategy. This should be better highlighted.
Rows 401, 403 and 405. As first word is better not include a number.
Rows 400-408. I read some percentages. I think that a bit improvement with stat comparison (with simple non parametric tests, as chi squared) should be useful to evidence if there are significant differences. See also Tab. 2. The discussion was largely focused on the differences between approaches (in percentages), with the field based method resulting the most effective. In this regard, it could be very useful add some inferential statistics, hghlighting as this effectiveness is significant.
Add the role of anonymous reviewers and Editors in the Acknowledments.
Have a nice work.
Author Response
85-86. Here the authors introduced the problem solving as concept. I think that they should refer to some seminal references about the approaches and techniques in this regard. For example, about conservation and sustainability, see ‘ Toward a new generation of effective problem solvers and project-oriented applied ecologists. Web Ecology, 20(1), 11-17, 2020 and ‘ Unifying the trans-disciplinary arsenal of project management tools in a single logical framework: Further suggestion for IUCN project cycle development. Journal for Nature Conservation, 41, 63-72.’ More particularly in this last paper it has been defined the different steps of a strategy (context, planning, process, monitoring). I think that these steps should be better defined in your paper. For example, Fig 1 refers to a classical ‘situation analysis’ of a project. However a large part of the paper can be considered an interesting ‘context analysis’ aimed to develop a strategy. This should be better highlighted. | Text added on lines 91-93: "This paper documents a well-tested field-based method for 91assessing trailbridge need, as well as the first three phasesof development and testing for remote 92methods to assess trailbridge need in the rural context." |
Rows 401, 403 and 405. As first word is better not include a number. | Language amended to not begin sentences with a number- now lines 423-436. "Cable-suspension bridges would be required at 38 percent of identified sites. 38 percent of identified sites required a cable suspension bridge in order to provide year-round access. In Liberia, all site validation was conducted in person due to limited telecommunications infrastructure. Of the sites identified via the field-based method, 13 percent of resulted in a location where an all-weather crossing (usually a vehicular bridge) had been constructed or was invalid, meaning there was no barrier along a road or walking path.13 percent of sites identified via the field-based method resulted in a location where an all-weather crossing (usually a vehicular bridge) had been constructed or was invalid, meaning there was no barrier along a road or walking path. Of the randomly selected sites, 82 percent needed a trailbridge (P<.001), and extrapolated to the full area of the two counties, we estimate a total of 867 trailbridge sites are needed in total, or approximately one trailbridge needed per 12.7 square kilometers.82 percent of the randomly selected sites were in need of a trailbridge, and extrapolated to the full area of the two counties, we estimate a total of 867 trailbridge sites are needed in total, or approximately one trailbridge needed per 12.7 square kilometers." |
I read some percentages. I think that a bit improvement with stat comparison (with simple non parametric tests, as chi squared) should be useful to evidence if there are significant differences. See also Tab. 2. The discussion was largely focused on the differences between approaches (in percentages), with the field based method resulting the most effective. In this regard, it could be very useful add some inferential statistics, hghlighting as this effectiveness is significant. | Chi square tests completed and added to line 494, table 2, table 3, and table 4 |
Add the role of anonymous reviewers and Editors in the Acknowledments. | Narrative added lines 623-625 - "Alissa Davis for her editing and narrative support; Joe McGlinchy at University of Colorado Boulder’s Earth Lab for his thoughtful comments; and the anonymous reviewers and editor for their careful reading and thoughtful comments on an earlier manuscript." |
Reviewer 2 Report
Overall, is an interesting paper, as it presents and details the field-based methods that could be used in the same area of interest by other countries that face the same kind of problems:
Somme suggestions for authors:
- Present clearly the aim of your study/research
- At Section 2 - Specify the research method and research tool used to conduct the study - e.g., is a case study? is this a content analysis based a documents provided by Bridges to Prosperity (B2P)? Are there some research questions? working hypothesis?
- Add the bibliographic source to each figure and table
- At Figure 8: Example of a map created during a community focus group, who has the ownership of this map?
- Interpret the results in in perspective of previous/similar studies/researches and of the working hypotheses.
studies/researches - are the results in accordance with the results of other studies on the topic?
- Specify if the ethics principles are totally respected in this study (especially related with the use/the property of the data analyzed in the study)
- Highlight the usefulness of this research
- Improve the key-words list ( is a case study on Liberia and Rwanda, than we suggest to add Liberia and Rwanda as key-words on the list)
- Check the style of citing the references!!!!See the Reference List and Citations Guide.
Author Response
Present clearly the aim of your study/research | Narrative added lines 54-57 - "The aims of this study were to understand the extent of the need for trailbridges in rural Rwanda and two counties of rural Liberia, and determine if simple remote methods of bridge site identification are viable and comparable to existing field-based methods." |
At Section 2 - Specify the research method and research tool used to conduct the study - e.g., is a case study? is this a content analysis based a documents provided by Bridges to Prosperity (B2P)? Are there some research questions? working hypothesis? | Narrative added lines 91-93 - "This paper documents a well-tested field-based method for assessing trailbridge need, as well as the first three phases of development and testing for remote methods to assess trailbridge need in the rural context." |
Add the bibliographic source to each figure and table | Added reference to QGIS - line 182 |
At Figure 8: Example of a map created during a community focus group, who has the ownership of this map? | The authors created this map so are the owners of the map |
Interpret the results in in perspective of previous/similar studies/researches and of the working hypotheses. studies/researches - are the results in accordance with the results of other studies on the topic? |
Narrative added lines 564-568 - "In general, this work builds on the usefulness of the Rural Access Index, a measure developed by the World Bank to evaluate transportation mobility, by evaluating the prevalence of a specific transportation barrier, and deploying rapidly developing technologies to generate results more efficiently than traditional methods [1,15]." |
Specify if the ethics principles are totally respected in this study (especially related with the use/the property of the data analyzed in the study) | Narrative added lines 133-138 - "Participation in focus groups was not compulsory. Assessors made it clear that project selection was ultimately up to the local government and that focus group participation would not influence the likelihood of a bridge being built at the identified site. Personally-identifiable information (including photos, names, or other personal information) information was not collected. All work was conducted in accordance with B2P’s programmatic objectives, and in line with established government partnerships." |
Highlight the usefulness of this research | Narrative added lines 593-596 - "This work underscores the need for trailbridges in rural regions, and the degree to trailbridges and other rural transportation infrastructure are both under-resourced and poorly understood, in addition to cataloguing the need for trailbridges in Rwanda and a portion of Liberia, and illuminating well-tested field methods for rural access assessments." |
Improve the key-words list ( is a case study on Liberia and Rwanda, than we suggest to add Liberia and Rwanda as key-words on the list) | Added "Rwanda; Liberia; rural access; GIS" to key words list |
Check the style of citing the references!!!!See the Reference List and Citations Guide | References have all been edited |
Reviewer 3 Report
the manuscript is centred on a theme of worldwide interest .It is considered appropriate to include a flowchart in the introduction explaining the theme of the manuscript .SI requires that the full name of each acronym used be included for a better understanding of the text
It is requested to include the limitations of the research and future steps of investigation and analysis.
It is necessary to insert fig.1 in high resolution with a text with the same font of the manuscript .
it is necessary to insert the source of the figures and maps used
In table 1 it is necessary to insert the units of measurement
Figure 5 must be preceded by a broader description.
Figure 8 is difficult to understand
It is necessary to insert more bibliographical references to accompany each paragraph.
Author Response
the manuscript is centred on a theme of worldwide interest .It is considered appropriate to include a flowchart in the introduction explaining the theme of the manuscript . | This comment is not well understood. Can the reviewer provide additional guidance on this comment? |
SI requires that the full name of each acronym used be included for a better understanding of the text | Full name of European Space Agency (ESA) added line 213 |
It is requested to include the limitations of the research and future steps of investigation and analysis. | Narrative added line 565-566 - "This work is limited in that it does not describe transportation barriers rivers that may exist between communities and essential destinations." |
It is necessary to insert fig.1 in high resolution with a text with the same font of the manuscript . | Figure 1 was updated with a high resolution image and the caption font was updated |
it is necessary to insert the source of the figures and maps used | QGIS citation added - line 182 |
In table 1 it is necessary to insert the units of measurement | Caption text edited to include units of measure in table 1 - "number of individuals and land cover" |
Figure 5 must be preceded by a broader description. | Narrative added lines 193-194 - "Figure 5 illustrates the two-kilometer radius created around a single site, and each discrete layer of data for which statistics were calculated within the radius." |
Figure 8 is difficult to understand | Caption text edited to clarify the figure - "Example of a map created during a community focus group; maps were created by B2P field assessors during focus groups. Numbers at points represent communities or destinations, which were logged and numbered, and in some cases, visited and documented" |
It is necessary to insert more bibliographical references to accompany each paragraph. | Reference 8 and 11 added |
Round 2
Reviewer 2 Report
The authors made improvements according the recommendations.
Author Response
No additional revisions requested
Reviewer 3 Report
the manuscript still has typos and grammatical errors.
It is necessary to insert an explanatory text between the flowchart and the following images that should include the source of the image (google, by author, openstreet map ...) but also a legend or colors that explain the differences between the images inserted.
If these have been inserted as a successive zoom of the same area should be specified in the text.
I do not think it appropriate that the manuscript has pages devoted only to images without a text that explains the real presence and usefulness in the text.
1)Campisi, T., Ignaccolo, M., Inturri, G., Tesoriere, G., & Torrisi, V. (2020, July). The Growing Urban Accessibility: A Model to Measure the Car Sharing Effectiveness Based on Parking Distances. In International Conference on Computational Science and Its Applications (pp. 629-644). Springer, Cham.
2) Koenig, J. G. (1980). Indicators of urban accessibility: theory and application. Transportation, 9(2), 145-172.
3) Mrak, I., Campisi, T., Tesoriere, G., Canale, A., & Cindrić, M. (2019, December). The role of urban and social factors in the accessibility of urban areas for people with motor and visual disabilities. In AIP Conference Proceedings (Vol. 2186, No. 1, p. 160008). AIP Publishing LLC.
4)Bhat, C., Handy, S., Kockelman, K., Mahmassani, H., Chen, Q., & Weston, L. (2000). Urban accessibility index: literature review. Austin: Texas Department of Transportation.
5)Rahman, M. T., & Nahiduzzaman, K. (2019). Examining the walking accessibility, willingness, and travel conditions of residents in Saudi cities. International journal of environmental research and public health, 16(4), 545.
Author Response
Comment from Reviewer | Response |
Reviewer 3 | |
the manuscript still has typos and grammatical errors. | |
It is necessary to insert an explanatory text between the flowchart and the following images that should include the source of the image (google, by author, openstreet map ...) but also a legend or colors that explain the differences between the images inserted. | Additional details and citations added to figures 2, 3, 4, 5, 6, 7, & 8 |
If these have been inserted as a successive zoom of the same area should be specified in the text. | Lines 207-208 added: "Figure 5 illustrates the two-kilometer radius created around a single site, and each discrete layer of data for which statistics were calculated within the radius." Additional details added to Figures 1-12. |
I do not think it appropriate that the manuscript has pages devoted only to images without a text that explains the real presence and usefulness in the text. | Descriptions of images were added to Figure 10, "A typical crossing in Rwanda where a bridge was requested is shown in Figure 10," Figure 11, "Figure 11 shows a typical crossing in Liberia where a bridge was requested," and Figure 12, "An example trailbridge which could be constructed to replace the typical timber crossings in Rwanda and Liberia is shown in Figure 12." |
1)Campisi, T., Ignaccolo, M., Inturri, G., Tesoriere, G., & Torrisi, V. (2020, July). The Growing Urban Accessibility: A Model to Measure the Car Sharing Effectiveness Based on Parking Distances. In International Conference on Computational Science and Its Applications (pp. 629-644). Springer, Cham. 2) Koenig, J. G. (1980). Indicators of urban accessibility: theory and application. Transportation, 9(2), 145-172. 3) Mrak, I., Campisi, T., Tesoriere, G., Canale, A., & Cindrić, M. (2019, December). The role of urban and social factors in the accessibility of urban areas for people with motor and visual disabilities. In AIP Conference Proceedings (Vol. 2186, No. 1, p. 160008). AIP Publishing LLC. 4)Bhat, C., Handy, S., Kockelman, K., Mahmassani, H., Chen, Q., & Weston, L. (2000). Urban accessibility index: literature review. Austin: Texas Department of Transportation. 5)Rahman, M. T., & Nahiduzzaman, K. (2019). Examining the walking accessibility, willingness, and travel conditions of residents in Saudi cities. International journal of environmental research and public health, 16(4), 545. |
A contextualization of the methods presented in the study, as well as relevant references, was added to the manuscript, in rows 91-102: "While methods for assessing pedestrian transportation access in environments have been extensively documented [11 – 13], there are significant gaps in the assessment of transportation access in remote and rural environments, particularly for those traveling by foot or motorcycle. The World Bank’s Rural Access Index is a valuable metric for estimating rural access in general, but does not provide guidance for identifying specific transportation barriers faced by rural communities. Work conducted by the Overseas Development Institute has demonstrated the relationship between isolation and poverty, and catalogued the myriad dimensions of isolation that influence a community’s ability to access essential services, but incorporates distance and the extent of road networks as important variables, and does not attempt to assess the specific transportation barriers that may be encountered on trail networks [14]. This paper documents a well-tested field-based method for assessing trailbridge need, as well as the first three phases of development and testing for remote methods to assess trailbridge need in the rural context." |